# LAMPO: Large Language Models as Preference Machines for Few-shot Ordinal Classification

**Zhen Qin, Junru Wu, Jiaming Shen, Tianqi Liu, Xuanhui Wang**
Google
{zhenqin,junru,jmshen,tianqiliu,xuanhui}@google.com

## Abstract

We introduce LAMPO, a novel paradigm that leverages Large Language Models (LLMs) for solving few-shot multi-class ordinal classification tasks. Unlike conventional methods, which concatenate all demonstration examples with the test instance and prompt LLMs to produce the pointwise prediction, our framework uses the LLM as a *preference machine* that makes a relative comparative decision between the test instance and each demonstration. A self-supervised method is then introduced to aggregate these binary comparisons into the final ordinal decision. LAMPO addresses several limitations inherent in previous methods, including context length constraints, ordering biases, and challenges associated with absolute point-wise estimation. Extensive experiments on seven public datasets demonstrate LAMPO's remarkably competitive performance across a diverse spectrum of applications (e.g., movie review analysis and hate speech detection). Notably, in certain applications, the improvement can be substantial, exceeding 20% in an absolute term. Moreover, we believe LAMPO represents an interesting addition to the non-parametric application layered on top of LLMs, as it supports black-box LLMs without necessitating the outputting of LLM's internal states (e.g., embeddings), as seen in previous approaches.

## 1 Introduction

In-Context Learning (ICL) with few-shot demonstrations, also known as few-shot prompting, is a prominent approach for multi-class classification using Large Language Models (LLMs) (Brown et al., 2020). ICL's remarkable performance and training-free nature, combined with off-the-shelf LLMs (e.g., GPT-4 (OpenAI, 2023) and PaLM2 (Google et al., 2023)), has significantly reduced the model adaption costs for new tasks and facilitated the concept of language-model-as-a-service (Sun et al., 2022). Despite these benefits, recent studies indicate that few-shot prompting can sometimes produce unreliable predictions and maybe not always outperform zero-shot prompting (Zhang et al., 2023). To understand these phenomena, extensive research has been conducted, leading to the following insightful observations:

**Restrictive Context Length Limit.** Despite extensive research efforts (Ratner et al., 2023; Press et al., 2021), the input context length limitation remains a major challenge in applying LLMs in many practical scenarios, especially given limited access to advanced commercial systems such as Reid et al. (2024). This constraint limits the number of demonstrations that can be included in a prompt, adversely affecting LLM performance, especially in challenging tasks with a large label space or long example texts. Also note that supporting long contexts does not mean the models necessarily perform well (Li et al., 2024) due to issues such as ordering bias, discussed next.

**Destructive Ordering Bias.** Recent studies (Liu et al., 2023; Lu et al., 2022) have identified a pronounced ordering bias in input prompts for LLMs. For standard few-shot multi-class classification tasks, the arrangement of demonstrations within the prompt can drastically affect the LLM performance, ranging from near chance to state-of-the-art levels. Addressing this issue is inherently complex due to the exponentially increasing combinations of demonstration orders as the number of examples grows.

**Challenges of Pointwise Estimation.** In many applications, pointwise estimation is notably more complex compared to comparative estimation. For instance, resolving pointwise relevance estimation addresses relevance ranking, but not conversely, a fundamental observation underpinning the learning-to-rank domain (Liu, 2009). Similarly, in fine-grained sentiment classification, LLMs may struggle to differentiate subtle sentiment nuances, such as between "very positive" and "positive", especially when given limited demonstrations in unfamiliar domains, or facing instructions not covered in their pre-training or instruction fine-tuning phases.

This study focuses on ordinal classification (Gutiérrez et al., 2016), an important subset of multi-class classification problem where the labels possess a natural order. Ordinal classification has extensive applications such as customer reviews analysis (Zhang et al., 2022), relevance rating in web search (Thomas et al., 2023), and opinion monitoring in social media (Barbieri et al., 2020). We propose to use instruction-following LLMs as *preference machines* to make pairwise comparisons between the test instance and each demonstration. In this paradigm, LLMs make *comparative*, rather than *absolute*, judgments in the ordinal label space, and we propose an unsupervised aggregation method to convert these comparison outcomes into the final ordinal prediction.

We present LAMPO, a novel framework that leverages LArge language Models as Preference machines for Ordinal classification. LAMPO implicitly addresses the above discussed ICL methods' limitations from three perspectives. First, it avoids packing all demonstrations in a single prompt and thus enables LLMs to leverage an arbitrary number of demonstrations. Second, it includes only two examples (one demonstration and one test instance) in each prompt and treats all prompts independently during the aggregation stage, which substantially mitigates the demonstration ordering bias in LLMs. Third, LAMPO employs the LLM as a preference machine and thus transforms the difficult pointwise estimation problem into a more manageable pairwise comparison problem.

The advantages of LAMPO are demonstrated by its competitive performance across seven diverse datasets, compatibility with both open-sourced (i.e., Flan-T5) and black-box LLMs (i.e., PaLM2), and reliance solely on binary generative decisions. This broad versatility makes it adaptable to various black-box LLMs, including those API-based LLMs with restricted access to internal states (e.g., embeddings) (Zhao et al., 2021). However, it is essential to acknowledge that our paradigm entails an increased number of LLM API calls (though with shorter sequences) and is only applicable to instruction-following LLMs, which we elaborate upon in Section 6.

Our contributions are summarized as follows:

- We introduce the LAMPO framework for the important ordinal classification problem. This framework effectively addresses several issues in the existing ICL paradigms.
- We present a pipeline that operates without the need for internal logits or embeddings from LLMs, and does not rely on an additional development dataset.
- We conduct extensive experiments on seven challenging datasets, covering the analysis of positivity, aspect-based sentiment, hatefulness, irony, and offensiveness of texts. LAMPO consistently demonstrates competitive performance when employed with various LLMs, where the improvements over state-of-the-art ICL approaches can be substantial.

## 2    Preliminary

In this section, we first introduce the original ICL method for few-shot ordinal classification with basic notations. Then, we discuss two representative ICL methods, which will shed lights on the difference between our work and the existing literature.

### 2.1    In-Context Learning (ICL) for Few-shot Ordinal Classification

ICL is a paradigm that allows LLMs to learn tasks given only a few demonstration examples. The standard $m$-way multi-class classification aims to assign the test input text $x$ to one candidate answer $y$ in the label space $Y = \{Y_0, ..., Y_{m-1}\}$. This paper focuses on ordinal

classification where there exists a natural order of the labels. Take the sentiment analysis task as an example, $Y_0$ denotes "very negative" and $Y_{m-1}$ denotes "very positive".

A large language model $M$ takes the candidate answer with the maximum score as the prediction conditioning a task instruction $I$ and a demonstration set $C$ which includes $k$ demonstration examples for each class[1]. Namely, we have $C = \{(x_i, y_i)|_{i=1}^{mk}\}$, where $(x_i, y_i)$ denotes one demonstration and there are a total of $mk$ ones. The likelihood of a candidate answer $Y_j$ could be represented by a scoring function $f$ of the whole input sequence with the LLM $M$:

$$P(Y_j|x) \triangleq f_M(Y_j, I, C, x). \tag{1}$$

The final predicted label $\hat{y}$ is the candidate answer with the highest probability:

$$\hat{y} = \arg\max_{Y_j \in Y} P(Y_j|x). \tag{2}$$

The scoring function $f$ estimates how possible the current answer is given the demonstration and the query text. For black-box generation-only LLMs, the most probable label string is expected to be generated directly.

## 2.2 Contextual Calibration (CC)

Existing work (Zhao et al., 2021; Fei et al., 2023) found that ICL is sensitive to the quality and ordering of demonstrations, leading to the calibration issue where the model biases towards a certain answer regardless of the input $x$. To mitigate this issue, contextual calibration (CC) (Zhao et al., 2021) first estimates the bias towards each answer by asking for the prediction probabilities $\hat{p}_{cf}$ of a content-free (cf) input such as "N/A", and then uses them to correct the prediction probabilities $\hat{p}_x$ ($[P(Y_1|x), ..., P(Y_m|x)]^T$) as

$$\hat{p}_x^{new} = diag(\hat{p}_{cf})^{-1}\hat{p}_x. \tag{3}$$

There are two major limitations of CC. First, CC requires LLM's internal states (i.e., probability logits) thus is not applicable to many black-box LLMs. Second, CC implicitly assumes the context-free input "N/A" can catch the deviation from a *uniform* prediction of the labels. This assumption might be less problematic in simpler classification scenarios, like binary sentiment analysis (positive vs negative). However, it does not hold for many real-world applications. For instance, in fine-grained classification tasks that include a "neutral" label, the "N/A" input likely has a non-trivial probability mass, leading to questionable calibration. Similarly, "N/A" assigns a probability mass to labels such as "non-hate" in hatefulness detection, "non-offensive" in offensiveness detection, etc. Thus, the practical utility of CC in such complex scenarios is limited.

## 2.3 Global Entropy (GlobalE)

Lu et al. (2022) studied the impact of demonstration ordering given a fixed demonstration set and presented a GlobalE method to identify prompts of specific demonstration orderings that prevent the extremely unbalanced prediction issue[2]. Specifically, GlobalE first generates multiple sets of candidate contexts by sampling different orderings of the demonstrations. Then, for each candidate contexts set $C_m$, it constructs a *probing set* by sampling from the LLM: $(x_i', y_i') \sim P_M(\cdot|C_m)$. See more details of probing set construction in Appendix B. After that, GlobalE computes the predicted label $\hat{y}_i$ for sampled data point under each $C_m$ as follows:

$$\hat{y}_i = \arg\max_{Y_j \in Y} f_M(Y_j, I, C_m, x_i'). \tag{4}$$

Finally, it ranks all candidate contexts based on the category label entropy of the predictions of the probing set, and uses the top-ranked context for actual inference. This method is

---

[1]There's ambiguity for "$k$-shot" in the literature. In this paper "$k$-shot" refers to $k$ demonstrations per class, as in Lu et al. (2022); Zhang et al. (2023).

[2]Lu et al. (2022) also proposed a LocalE method that depends on internal states of LLMs and under-performs GlobalE in most cases. Thus, we omit this method in the paper.

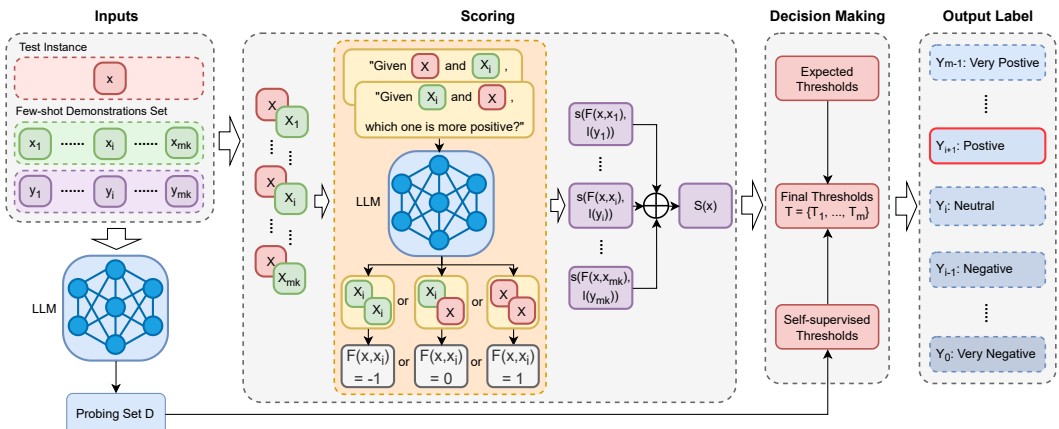

Figure 1: An illustration of our LAMPO on a m-way sentiment classification task. The scoring step compares test instance with every demonstration using LLM. The decision making step converts the score to an ordinal label by using thresholds learned offline.

effective under the mild assumption that the probing set should have a uniform distribution over the label space, since they were sampled using sampled $C_m$s, which should not have strong biases towards any labels globally.

GlobalE does not depend on LLM internal states and generally outperforms basic ICL in our experiments. However, it can still be limited by restrictive context length and difficulties of pointwise estimation. Also, given the combinatorial nature of possible orderings, candidate contexts $C_m$ can only be sampled and are unlikely optimal.

## 3 Scope

To address the ambiguity surrounding the term "ICL" (Dong et al., 2022), it is necessary to delineate the scope of this research. We focus on the practical "true" $k$-shot learning (Perez et al., 2021) for multi-class classification with off-the-shelf LLMs, where at most $k$ demonstrations are available for each class. This is distinct from previous studies that can retrieve a fixed number of demonstrations from a potentially large labeled dataset (Liu et al., 2021).

Furthermore, this work does not assume the presence of a labeled development set for tuning hyperparameters, as such an assumption would violate the aforementioned setting. This better aligns with the standard setting from the original ICL work (Brown et al., 2020) and several key subsequent ICL research (Zhao et al., 2021; Lu et al., 2022) discussed above. The comparison with methods involving test example dependent demonstration retrieval and LLM fine-tuning is beyond the scope of this paper.

## 4 LAMPO

We now describe our new paradigm LAMPO for few-shot ordinal classification, with the overall diagram shown in Figure 1. LAMPO consists of two steps: (1) a *scoring* step that uses the LLM to compute the comparative score between each test instance with each demonstration, and (2) a *decision making* step that converts those scores into ordinal labels by using offline learned thresholds without an additional labeled development set.

### 4.1 Scoring

We score each test instance $x$ as follows:

$$S(x) \triangleq \sum_{(x_i, y_i) \in C} s(F(x, x_i), l(y_i)). \tag{5}$$

The fundamental computation unit of LAMPO is denoted as $F(x, x_i)$, which involves LLM calls to *compare* input $x$ with each demonstration $x_i$ within the ordinal label space. For example, in hatefulness detection, the prompt can be "Given Passage A: $\{x\}$ and Passage B:

$\{x_i\}$, which one is more hateful?". For the exact prompts used in each dataset, please refer to Appendix C.

When computing $F(x, x_i)$, we can significantly mitigate the position bias by making two LLM calls with swapped orderings of $x$ and $x_i$ in the prompts, which is not feasible for existing ICL methods with combinatorial orderings. The resulting value of $F(x, x_i)$ can assume one of three outcomes: $F(x, x_i) = 1$ indicates $x$ is preferred (both LLM calls prefer $x$), $F(x, x_i) = -1$ signifies a preference for $x_i$ (both LLM calls prefer $x_i$), and $F(x, x_i) = 0$ denotes a tie (the two LLM calls yield conflicting or inconclusive results).

The local score $s$ for comparing $x$ and $x_i$ is derived from the comparison result $F(x, x_i)$ and label $y_i$. In line with established practices in ordinal classification literature, we use an integer number to represent each label $Y$, namely, $l(Y_j) = j$. This approach simplifies our scoring process as follows:

$$s(F(x, x_i), l(y_i)) = l(y_i) + F(x, x_i). \tag{6}$$

For example, if $x$ wins over $x_i$, it gets a local score of $l(y_i)$ plus 1 point. These local scores are added together as the final score $S(x)$ following Eq. 5.

## 4.2 Decision Making

Given the score $S(x)$, we need to convert it back into the ordinal label space, which can be reduced to the problem of identifying $m - 1$ thresholds $T = \{T_1, ..., T_{m-1}\}$, so that:

$$\hat{y} = \begin{cases} Y_{m-1} & \text{if } S(x) \geqslant T_{m-1} \\ Y_{m-2} & \text{if } T_{m-2} \leqslant S(x) < T_{m-1} \\ ... \\ Y_0 & \text{if } S(x) < T_1 \end{cases} \tag{7}$$

Note that if there exists a labeled development set, the search of thresholds would be easy. However, it will violate the "true" few-shot setting. We thus propose the following strategies to find $T$:

**Expected Thresholds.** Intuitively, imagine we have a test example with label value $Y_j$, we know the scores of comparing it with each demonstration from Eq. 6 if everything is noise-free. We can easily derive the *expected* score of an example of any label value $Y_j$:

$$\begin{aligned} S_j = \quad & \sum_{(x_i, y_i)|y_i < Y_j} (l(y_i) + 1) + \sum_{(x_i, y_i)|y_i = Y_j} \\ & l(y_i) + \sum_{(x_i, y_i)|y_i > Y_j} (l(y_i) - 1) \end{aligned} \tag{8}$$

This computation does not require any LLM calls since we just derive the expected score given the demonstration labels. Then the *expected threshold* $T_j$ is simply $(S_j + S_{j-1})/2$, and the other thresholds can be derived similarly. However, this is under the assumptions that the demonstration labels are noise free and pairwise comparisons are unbiased globally, which is hardly true in practice. Still, we can view this as a global *prior* of the thresholds.

**Self-supervised Thresholds.** Systematic bias can arise from both the demonstrations and LLM itself and the expected thresholds may not be optimal. We are thus interested in decision thresholds that are adaptive to the current demonstrations and LLM. Inspired by Lu et al. (2022), we leverage the probing set to rank candidate thresholds based on the global entropy of predicted labels. While we believe applying the probing set to our problem is a contribution, the probing set construction method itself is not our contribution, which is deferred to Appendix B. Different from Lu et al. (2022) where their candidate space is combinatorial in terms of the number of demonstrations, our search space is combinatorial in terms of the number of integer thresholds, which is usually small. Note that the LLM calls that compare each example in the probing set and demonstrations are only required once before the threshold search. The major caveat of this self-supervised approach is potentially large variance due to the limited number of demonstrations. We can treat these "self-supervised thresholds" as the *empirical observation*.

**The Mixture.** Given the pros and cons of the above two methods, we propose to use a mixture of above two strategies, which can be treated as a simple combination of prior and empirical thresholds. Since we do not assume a development set, we simply use the average of their corresponding thresholds without tuning the weights for all experiments.

### 4.3 A Running Example

Let's consider a 5-shot 3-way ordinal classification with labels {negative (0), neutral (1), positive (2)}. The expected thresholds would be {10, 20}, which can be derived as follows: a negative example is expected to have a score 5 in a noise-free setting (it will tie with the 5 negative demonstrations, and lose to the other 5 neutral and 5 positive demonstrations, thus $0 + 0 + 5$), a neutral example with score $15(5 + 5 + 5)$, and a positive example with score $25(5 + 10 + 10)$, thus the expected thresholds are $10 = (5 + 15)/2$ and $20 = (15 + 25)/2$.

Furthermore, let's assume the self-supervised thresholds are {12, 21}, then our final thresholds would be {11, 20.5}. Now given a test instance $x$, we do comparisons using LLM with each of the 15 demonstrations and get a score $S(x) = 16$ according to Eq. 5, then the final predicted label of $x$ will be "neutral".

## 5 Experiments

### 5.1 Datasets

We use 7 challenging ordinal classification datasets, i.e., Twitter, SST-5, Hate, Yelp-5, Lap14, Offensive, and Irony. We follow the publicly hosted setting[3] where each dataset provides 3 seeds of demonstrations in each configuration, i.e., 5-shot and 10-shot, which are used to compute mean and standard deviation of the concerned metrics on the test set. *The same publicly hosted setting is used for all compared methods for a fair and reproducible comparison.* The datasets are summarized in Table 3 in Appendix A, including their tasks, metrics, and label spaces.

### 5.2 Baselines and Model Configurations

For few-shot LLM-based methods, we compare LAMPO with ICL, CC, and GlobalE, which were discussed in Section 2. For few-shot LLM-based methods, we use the black-box PaLM2-S model (text-Bison[4]) and the white-box Flan-T5-XXL (Chung et al., 2022). We use the more general generation mode, that does not request any internal states of LLMs, for ICL, GlobalE, and LAMPO. We use the scoring mode to get the required internal logit values of label texts with Flan-T5-XXL for CC.

Note that due to potential data leakage issues of modern LLMs (Zhou et al., 2023) (see more discussions in Section 6.4), we will focus on comparing different methods under the same LLM, and only briefly compare trends of performance across different LLMs.

### 5.3 Experimental Results

This section describes our experimental results, which are shown in Table 1 and Table 2. We discuss datasets in groups due to their diverse settings and behaviors.

**Results on Twitter.** LAMPO exhibits the best performance on PaLM2-S and Flan-T5-XXL. Contrastingly, CC is ineffective due to the presence of a "neutral" class that violates its calibration assumptions. Furthermore, it's worth noting that LAMPO is the sole method capable of handling 10-shot scenarios on this dataset using the concerned popular LLMs. The slightly lower performance in this 10-shot setting compared to the 5-shot setting across both LLMs highlights the importance of the demonstrations' quality, as other factors, including ordering bias, have been largely mitigated.

---

[3]https://github.com/DAMO-NLP-SG/LLM-Sentiment/tree/master/data
[4]https://cloud.google.com/vertex-ai/docs/generative-ai/model-reference/text

Table 1: Experimental results on PaLM2-S models. "NA" means "Not Applicable" for ICL and GlobalE denotes infeasible experiments due to limited sequence length. CC is marked as "NA" as it requires outputting logits of each label, which is not available for black-box LLMs such as PaLM2-S. The best performing method is bolded for each row.

| Dataset | Metric | Config | Few-shot LLM-based Methods with PaLM2-S | | | |
| --- | --- | --- | --- | --- | --- | --- |
| | | | ICL (Brown et al., 2020) | CC (Zhao et al., 2021) | GlobalE (Lu et al., 2022) | LAMPO (Ours) |
| Twitter | Acc | 5-shot | $65.13_{0.25}$ | NA | $65.00_{0.49}$ | $\mathbf{66.87}_{0.52}$ |
| | | 10-shot | NA | NA | NA | $\mathbf{65.87}_{0.52}$ |
| SST-5 | Acc | 5-shot | $54.8_{0.85}$ | NA | $54.53_{0.50}$ | $\mathbf{55.4}_{1.41}$ |
| | | 10-shot | NA | NA | NA | $\mathbf{56.00}_{1.50}$ |
| Yelp-5 | Acc | 5-shot | NA | NA | NA | $\mathbf{50.67}_{1.64}$ |
| | | 10-shot | NA | NA | NA | $\mathbf{51.13}_{0.19}$ |
| Lap14 | Acc | 5-shot | $75.07_{1.00}$ | NA | $76.53_{1.23}$ | $\mathbf{78.13}_{0.66}$ |
| | | 10-shot | NA | NA | NA | $\mathbf{78.73}_{0.66}$ |
| Hate | macro_f1 | 5-shot | $48.81_{2.41}$ | NA | $49.94_{2.67}$ | $\mathbf{68.88}_{0.38}$ |
| | | 10-shot | $43.18_{5.32}$ | NA | $46.22_{4.39}$ | $\mathbf{71.43}_{1.92}$ |
| Offensive | macro_f1 | 5-shot | $80.23_{2.18}$ | NA | $\mathbf{82.38}_{0.48}$ | $80.82_{0.39}$ |
| | | 10-shot | $76.99_{4.69}$ | NA | $76.90_{6.53}$ | $\mathbf{81.10}_{0.47}$ |
| Irony | f1(irony) | 5-shot | $78.93_{2.95}$ | NA | $79.60_{2.47}$ | $\mathbf{89.61}_{1.13}$ |
| | | 10-shot | $78.40_{1.98}$ | NA | $80.65_{1.60}$ | $\mathbf{89.53}_{0.58}$ |

Table 2: Experimental results on Flan-T5-XXL models. "NA" means "Not Applicable" for ICL, CC, and GlobalE denotes infeasible experiments due to limited sequence length. The best performing method is bolded for each row.

| Dataset | Metric | Config | Few-shot LLM-based Methods with Flan-T5-XXL | | | |
| --- | --- | --- | --- | --- | --- | --- |
| | | | ICL (Brown et al., 2020) | CC (Zhao et al., 2021) | GlobalE (Lu et al., 2022) | LAMPO (Ours) |
| Twitter | Acc | 5-shot | $59.00_{1.42}$ | $50.53_{0.10}$ | $63.67_{0.57}$ | $\mathbf{64.55}_{0.21}$ |
| | | 10-shot | NA | NA | NA | $\mathbf{63.88}_{0.61}$ |
| SST-5 | Acc | 5-shot | $33.13_{3.21}$ | $\mathbf{53.4}_{1.14}$ | $37.07_{2.47}$ | $38.73_{2.86}$ |
| | | 10-shot | NA | NA | NA | $\mathbf{39.47}_{1.96}$ |
| Yelp-5 | Acc | 5-shot | NA | NA | NA | $\mathbf{32.93}_{2.23}$ |
| | | 10-shot | NA | NA | NA | $\mathbf{33.07}_{1.64}$ |
| Lap14 | Acc | 5-shot | $76.27_{0.62}$ | $71.93_{0.25}$ | $77.23_{0.25}$ | $\mathbf{77.47}_{0.52}$ |
| | | 10-shot | NA | NA | NA | $\mathbf{76.93}_{0.62}$ |
| Hate | macro_f1 | 5-shot | $54.53_{1.55}$ | $30.68_{0.61}$ | $58.84_{0.76}$ | $\mathbf{61.57}_{0.81}$ |
| | | 10-shot | $48.77_{9.86}$ | $30.59_{0.49}$ | $54.43_{13.05}$ | $\mathbf{61.73}_{2.37}$ |
| Offensive | macro_f1 | 5-shot | $80.63_{1.38}$ | $59.80_{4.06}$ | $\mathbf{80.71}_{0.68}$ | $78.26_{0.08}$ |
| | | 10-shot | $60.61_{27.41}$ | $60.60_{13.25}$ | $60.33_{27.19}$ | $\mathbf{78.42}_{0.20}$ |
| Irony | f1(irony) | 5-shot | $\mathbf{87.85}_{0.17}$ | $63.63_{0.68}$ | $87.66_{0.83}$ | $86.37_{0.66}$ |
| | | 10-shot | $86.72_{0.57}$ | $64.43_{2.39}$ | $\mathbf{88.11}_{0.61}$ | $86.83_{1.10}$ |

**Results on SST-5.** LAMPO works well on this highly challenging dataset with 5-level labels, outperforming ICL and GlobalE with both LLMs. 10-shot also consistently outperforms 5-shot with LAMPO, while it is infeasible to perform 10-shot with other methods. CC with 5-shot performs well with Flan-T5-XXL, as it avoids generation errors on this challenging (for Flan-T5-XXL) dataset. See more discussions in the summary of results below.

**Results on Yelp-5.** Results on Yelp-5 mainly shows the capability of LAMPO - no other methods are feasible given the long example texts. Also, 10-shot works better than 5-shot on both LLMs for LAMPO. LAMPO allows us to find that there still exists a gap between LAMPO and supervised methods, so the community is suggested to focus on challenging datasets like Yelp-5 for future work.

**Results on Lap14.** LAMPO works well on this dataset, with the best performance on both PaLM2-S and Flan-T5-XXL, showing LAMPO is compatible with the challenging aspect-based sentiment analysis task. CC works poorly since there is a "neutral" class.

**Results on Hate.** We get very strong results of LAMPO on PaLM2-S, where the absolute performance gain can be larger than 20%. LAMPO also generates the best performance with Flan-T5-XXL, though the gap is smaller. We hypothesize this is because PaLM2-S was not exposed to similar datasets, especially with regard to their label space, during

its pre-training and instruction fine-tuning. In contrast, LAMPO handles a more general "hateful" concept that PaLM2-S is familiar with, regardless of whether it has encountered the exact label space before or not. CC again works poorly because the probability mass will be assigned to "non-hate" for the "N/A" input.

**Results on Offensive and Irony.** LAMPO is competitive and robust for these two datasets on both LLMs. One noticeable benefit of LAMPO is its robustness across different shots - the performance of other methods on Offensive actually drops with 10 shots, but LAMPO is robust and generally improves performance with more shots. CC does not work as the probability mass will be concentrated on "non-offensive" and "non_irony".

**Summary of Results.** On the seven tasks across a broad range of domains we find that:

(1) LAMPO is very competitive and works the best in most cases. Importantly, LAMPO is the most *robust* method, while other methods have their respective failure patterns. In general, LAMPO shows better comparative performance with respect to other methods on PaLM2-S than Flan-T5-XXL, indicating that more advanced LLMs tend to do comparisons better across various domains.

(2) LAMPO enables popular LLMs the capability to perform competitive ordinal classification in various settings, including black-box or white-box LLMs, number of shots, and on tasks where the LLM may not have strong prior knowledge of the label space.

(3) The calibration assumption of CC is too strong and this impairs its robustness. Despite its strong performance on SST-5 with Flan-T5-XXL, it can not be used with black-box LLMs and it hurts performance on most datasets due to violation of its calibration assumption.

(4) GlobalE outperforms the ICL baseline in most cases, but it is generally less effective than LAMPO, possibly because the ordering bias is intrinsically difficult to address due to the combinatorial nature of possible orderings.

# 6 Discussion and Limitation

## 6.1 When do we expect LAMPO to be most valuable?

LAMPO shows competitive performance across different domains. One most prominent improvement LAMPO got is on the hatefulness detection dataset, where we hypothesize that PaLM2-S does not have a strong prior understanding of the task, especially concerning the "non-hate" label. However, LAMPO mainly leverages more general knowledge such as "which passage is more hateful?" and *does not use the actual label text* until the very last decision making step. Thus, we believe LAMPO is particularly valuable when the LLM lacks a strong prior understanding of the task's label space.

## 6.2 Restriction to Instruction-following LLMs

Earlier research before the ubiquity of instruction-following LLMs on ICL mainly depends on text continuation. On the other hand, as hinted by the prompts in Appendix C, the comparison prompts used in LAMPO are instruction-following and are likely only applicable on instruction-following LLMs. This is one limitation of our framework. However, we argue the limitation may not be significant as instruction-following LLMs are becoming the norm.

## 6.3 Cost

We acknowledge that LAMPO does involve an increased number of LLM API calls compared to standard ICL methods. However, we note that: (1) The number of calls only linearly scales with the number of demonstrations, which is usually not big in the concerned "few-shot" setting. On the other hand, LAMPO's new capability allows for the potential utilization of an unlimited number of demonstrations, making the pursuit of more efficient methods a promising future research direction. (2) Each prompt is shorter, so each API call is faster

and less expensive than traditional ICL. (3) All comparison API calls can be paralleled, thus the latency can be lower than traditional ICL, given sufficient parallelism.

### 6.4 Data Leakage

It is possible that certain LLMs saw some datasets during their pre-training or instruction fine-tuning, which is a major concern of fair evaluation in the era of LLMs (Zhou et al., 2023). We mainly focused on comparing different methods using the same LLMs so the comparisons are fair. In fact, as most existing LLM pre-training and instruction tuning methods adopt the pointwise strategy, a potential data leakage will benefit them more, compared to our pairwise framework. Thus, the improvements achieved by LAMPO over these baselines further demonstrate its effectiveness.

### 6.5 Reproducibility

As noted, all datasets used in this paper are publicly available, and we did not change any of their demonstrations, labels, or any other content. The exact prompt templates we used are shown in Appendix C.

## 7 Related Work

**In-Context Learning.** We have discussed several popular ICL-based methods in Section 2, which is a prominent family of approaches that leverages LLM for few-shot multi-class classification. Another emerging paradigm, which has gained popularity recently, involves retrieving distinct demonstrations for each test instance (Rubin et al., 2021). These demonstrations are sourced from a potentially vast annotated demonstration pool, which is very different from the "true" few-shot setting studied in this work.

**Comparisons using LLMs.** We note that the "pairwise" or "comparative" paradigm is a classic concept in various domains. LLMs can serve as evaluators to compare preferences of generative model's **outputs** (Kocmi & Federmann, 2023; Vu et al., 2024; Yan et al., 2023), such as to determine which summary is better in text summarization, with many important applications such as language model alignment (Liu et al., 2024). In contrast, our approach compares **inputs** to derive an output. For web search ranking, given a query, pairs of candidate documents can be compared using LLMs in terms of their relevance to the query (Qin et al., 2024). This task significantly differs from ours, and it involves a quadratic increase in API calls with the number of documents under each query. In summary, existing work only share the very general idea of doing pairwise comparisons with very different problem setups where natural pairs are readily available. In contrast, our work pioneers the exploration of LLMs as preference machines for the important few-shot ordinal classification task and demonstrates LLMs' comparative capability in broader dimensions (i.e., traditional sentiment, hatefulness, offensiveness, irony, and aspect-based sentiment).

**"Non-parametric" application of LLMs.** Very recently, Xu et al. (2023) have shown a "nearest neigbor" style application of LLMs. We posit that our paradigm offers a compelling addition to this non-parametric perspective of LLM utilization, where existing research (Xu et al., 2023) hinges upon extracting embeddings that are not generally accessible, and neglects to explore the preference machine perspective of LLMs.

## 8 Conclusion

This paper introduces a simple yet novel paradigm, LAMPO, for the important $k$-shot multi-class ordinal classification problem with LLMs. LAMPO effectively addresses several inherent limitations of traditional ICL methods, and demonstrates strong performance on 7 publicly available datasets covering diverse topics. Moreover, LAMPO is versatile, compatible with both black-box and white-box LLMs, and capable of accommodating an arbitrary number of demonstrations.

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

# A    Dataset Descriptions

We show the descriptions of each dataset used in our evaluation in Table 3.

Table 3: A description of the datasets used in evaluations. The labels are ordered in the ordinal space

| Dataset | Label Space | Metric | Task |
|---------|-------------|--------|------|
| Twitter | negative, neutral, positive | Accuracy | Sentiment analysis on social media posts. |
| SST-5 | very negative, negative, neutral, positive, very positive | Accuracy | Sentiment analysis on movie reviews. |
| Yelp-5 | very negative, negative, neutral, positive, very positive | Accuracy | Sentiment analysis on customer reviews on businesses. |
| Lap14 | negative, neutral, positive | Accuracy | Aspect-based sentiment analysis. |
| Hate | non-hate, hate | Macro F1 | Hate speech detection. |
| Offensive | non-offensive, offensive | Macro F1 | Offensive language identification. |
| Irony | non_irony, irony | F1 of irony | Irony detection. |

# B    The Probing Set

We provide more details on the construction and use of the "probing set", with most contents below adopted from Lu et al. (2022). The key idea is to build a collection of examples that has a roughly uniform label distribution. Then given a candidate in the search space (a set of thresholds in LAMPO and a specific ordering of demonstrations in GlobalE), the predictions of the examples can be generated, and the best candidate should produce label predictions closest to the uniform distribution, measured by entropy.

To construct such a "probing set" without access to any additional data, Lu et al. (2022) proposes to directly sample from an LLM. Concretely, given a set of training / demonstration examples $S = (x_i, y_i), i = 1, ..., n$, we define a transformation $T$, mapping each example into natural language space. We use a simple transformation function $T$ such that $T(x_i, y_i) =$ input : $x_i$ type : $y_i$. This transforms each example into a standard format sentence, which linearises each element in the set into the natural language space defined as $S' = \{t_i\}, i = 1, ..., n$.

We then define a full permutation function group of $n$ training examples, $\mathcal{F} = \{f_m\}, m = 1, ..., n!$, where each function $f_m$ takes $S'$ as input and outputs $C_m$: the concatenation of a unique permutation. For each prompt candidate $C_m$, we then sample from the large language model $M$ to obtain the probing sequence $g_m \sim P_M(\cdot | C_m)$. We stop decoding from the language model upon generating the special end-of-sentence token defined by a template, or reach the generation length limit. The probing set construction method is illustrated in Figure 2, where the objective is to generate a probing set that shares a similar distribution to the training examples.

We run this sampling process for sampled prompt ordering permutations and extract probing examples from them, then gather extracted examples together to form the probing set $D$. Although the probing set contains predicted label for each sentence, there is no guarantee on the validity of these labels. Therefore, we discard them from the probing set as we are only interested in sampling probes from the language model corresponding to the input distribution.

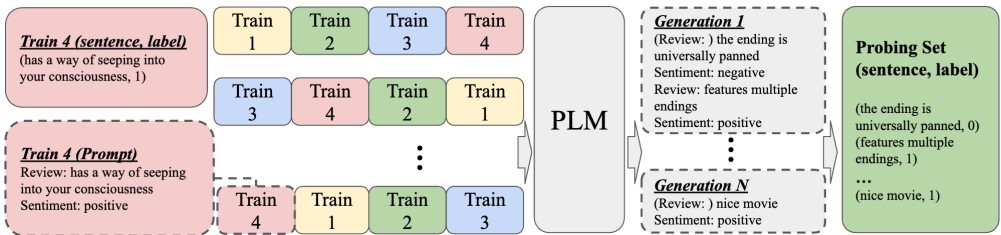

Figure 2: An illustration of probing set construction from Lu et al. (2022). "PLM" is the same as "LLM" in the context of this paper.

Once we have constructed a probing set for a given set of demonstrations, we can now use that probing set to identify the best possible prompt order as in Lu et al. (2022) and thresholds in LAMPO. In this work, we use PaLM2-S to generate 50 examples in each configuration, and the same probing set is used for LAMPO and GlobalE for a fair comparison.

## C    Prompt Templates

We show the prompt template used by LAMPO on each dataset. In general, we tried to avoid the usage of specific words such as "Tweet" or "Review", but use a more generic word "Passage". Also, we did not heavily tune the prompts on each dataset, some datasets even share the same prompt template. These demonstrate the generality and robustness of the LAMPO paradigm, and better performance should be achieved with better prompt engineering.

### C.1    Twitter Dataset

> Given two Passages, compare their sentiments with labels from ['negative', 'neutral', 'positive'].
>
> Passage A: {item$_1$}
>
> Passage B: {item$_2$}
>
> Which Passage is more positive in terms of its sentiment?
>
> Output Passage A or Passage B:

### C.2    SST-5 and Yelp-5 Datasets

> Given two Passages, compare their sentiments with labels from ['very negative', 'negative', 'neutral', 'positive', 'very positive'].
>
> Passage A: {item$_1$}
>
> Passage B: {item$_2$}
>
> Which Passage is more positive in terms of its sentiment?
>
> Output Passage A or Passage B:

### C.3 Lap14 Dataset

Given two Passages, compare their sentiments towards their respective aspects with labels from ['negative', 'neutral', 'positive'].

Passage A: {item$_1$} (sentiment towards {aspect$_1$}),

Passage B: {item$_2$} (sentiment towards {aspect$_2$})

Which Passage is more positive in terms of its sentiment towards its aspect?

Output Passage A or Passage B:

### C.4 Hate Dataset

Given two Passages, compare their sentiments with labels from ['non-hate', 'hate'].

Passage A: {item$_1$}

Passage B: {item$_2$}

Which Passage is more hateful in terms of its sentiment?

Output Passage A or Passage B:

### C.5 Offensive Dataset

Given two Passages, compare their sentiments with labels from ['non-offensive', 'offensive'].

Passage A: {item$_1$}

Passage B: {item$_2$}

Which Passage is more offensive in terms of its sentiment?

Output Passage A or Passage B:

### C.6 Irony Dataset

Given two Passages, compare their irony with labels from ['non_irony', 'irony'].

Passage A: {item$_1$}

Passage B: {item$_2$}

Which Passage is more ironic in terms of its sentiment?

Output Passage A or Passage B:

