# OpenReview forum: "LAMPO: Large Language Models as Preference Machines for Few-shot Ordinal Classification"
_colmweb.org/COLM/2024/Conference — COLM_

### Official Review · Reviewer_pYRD · 2024-05-10

**Rating:** 6
**Confidence:** 4
**Ethics Flag:** 1

**Summary:**

This paper introduces LAMPO, a new method for ordinal classification (e.g., sentiment analysis) using LLMs. The main idea is to use the LLM as a preference machine, making pairwise comparisons between a test instance with each demonstration example. This approach addresses limitations of prior ICL methods, such as context length constraints and ordering biases. LAMPO aggregates these pairwise comparisons into a final ordinal prediction using a self-supervised method. Experiments on several datasets show LAMPO's good performance with improvement over state-of-the-art ICL approaches.

**Reasons To Accept:**

- LAMPO presents a new approach for ordinal classification with LLMs, using them as preference machines rather than for direct point-wise prediction. This approach is innovative and could inspire further research in this direction.

- The paper demonstrates LAMPO's performance across diverse datasets, outperforming existing ICL methods in many cases. The significant improvements in some tasks highlight the potential impact of this approach.

- LAMPO is compatible with both open-source and black-box LLMs, making it widely applicable. It also doesn't require access to internal LLM states, which increases its flexibility.

**Reasons To Reject:**

- The blackbox LLM model used in the paper are not SOTA ones, making it less convincing on whether the proposed approach can still outperform other ICL approaches.

- The paper acknowledges that LAMPO requires more LLM API calls than traditional ICL. While it argues that the calls are shorter and can be parallelized, the overall cost implications is still a major concern given the pair-wise comparison.

- The evaluation is primarily focused on empirical performance. More in-depth analysis is recommended to understand why LAMPO works well and under what conditions it might fail.

---

> ### Author Rebuttal · Authors · 2024-05-30
>
> We appreciate your constructive feedback and advocate of our paper. Here we discuss some questions/concerns you raised.
>
> - The blackbox LLM model used in the paper are not SOTA ones, making it less convincing on whether the proposed approach can still outperform other ICL approaches.
>
> We acknowledge this comment. As discussed in the paper,  the two LLMs chosen are representative in terms of black-box vs white box, and vary in their sizes. Performance comparison is focused on comparing different methods using the same LLMs. PaLM2 is still popular at the time of writing the paper and available as a commercial API. Benchmarking on more LLMs is a meaningful effort.
>
>
> - The paper acknowledges that LAMPO requires more LLM API calls than traditional ICL. While it argues that the calls are shorter and can be parallelized, the overall cost implications is still a major concern given the pair-wise comparison.
>
> Yes, the number of API calls increases. As LAMPO is the first work along the concerned direction, we did not focus on cost optimization. Interesting future work can include smarter pair selection and we are excited about the possibilities.
>
>
> - The evaluation is primarily focused on empirical performance. More in-depth analysis is recommended to understand why LAMPO works well and under what conditions it might fail.
>
> Thank you for your suggestion. We discussed one key observation in section 6.1, where we observed that existing ICL methods can be brittle to label strings. As discussed in the paper, the major strength of LAMPO is its robustness and applicability across different settings. On the benchmark we tested, we did not find clear failure patterns of LAMPO.

---

> > ### Comment · Reviewer_pYRD · 2024-06-03
> >
> > Although my assessment remains the same after reviewing it, I appreciate you taking the time to respond.

---

### Official Review · Reviewer_cX41 · 2024-05-11

**Rating:** 8
**Confidence:** 4
**Ethics Flag:** 1

**Summary:**

This work studies multi-class ordinal classification (i.e., classification where there is an inherent ordering between the classes) using LLM prompting with demonstrations (i.e., in-context learning). Simply concatenating demonstrations for in-context learning has challenges, including (1) context length limitation, (2) ordering bias, and (3) pointwise estimation. The paper proposes LAMPO to address these challenges. LAMPO can be viewed as a prompting technique where each demonstration is compared against the test example separately, and the LLM judge is only required to make pairwise comparisons. Given $m$ demonstrations, the LLM judge performs $m$ comparisons (or $2m$ for symmetry) and then the outcomes are aggregated to obtain a final predicted class. The work considers a novel method to map from combined output into the final class. Experiments are conducted on a range of standard datasets (e.g., Twitter, SST-5, Yelp-5, etc), and the results demonstrate the effectiveness of LAMPO.

**Questions To Authors:**

Have you considered applying LAMPO (in your few-shot learning manner) to evaluate open-ended text generation tasks?

**Reasons To Accept:**

1. LAMPO is a simple and novel prompting idea, which can mitigate the three challenges
2. The results illustrate the effectiveness of the proposed approach on several datasets
3. The approach is applicable to black-box LLMs

**Reasons To Reject:**

1. The method requires many calls to LLMs (e.g., $2m$ call for each test example with $m$ demonstration).
2. The method could become less relevant as the context length of LLMs are becoming longer.

---

> ### Author Rebuttal · Authors · 2024-05-30
>
> We appreciate your constructive feedback and advocate of our paper. Here we discuss some questions/concerns you raised.
>
> - The method requires many calls to LLMs (e.g., 2𝑚  call for each test example with 𝑚 demonstration).
>
> Yes, the number of API calls increasing is a limitation of LAMPO at its current form and we fully acknowledge it (also in the paper). As LAMPO is the first work along the concerned direction (few-shot comparison-based ordinal classification), we did not focus on cost optimization. Interesting future work can include smarter pair selection and debiasing to save half of the calls.
>
>
> - The method could become less relevant as the context length of LLMs are becoming longer.
>
> We agree with you. We believe that LAMPO may stay relevant for some time since, even if long context LLMs may address the input length constraint, it may not address the other constraints (position bias, difficulty of pointwise estimation). In fact, it is unclear if the position bias will be more severe or not with longer context given the “lost in the middle” behavior commonly found in LLMs. Long context LLMs such as Gemini-1.5 are promising but further research might be needed to address possible pitfalls raised from such models before they become the default choice.
>
> - Have you considered applying LAMPO (in your few-shot learning manner) to evaluate open-ended text generation tasks?
>
> This is an interesting idea for an important problem, and can be a promising future research direction. We believe that the general idea of LAMPO (preference + decision making) can be applied to a wide range of ordinal classification / ranking problems, which is ubiquitous in practice. For example, as you suggested, we may want to measure the safety (not-safe vs safe, or more fine-grained safety levels than the two options) of a text generation, and may follow LAMPO to improve the prediction accuracy given some  labeled instances. These can be impactful topics to explore.

---

> > ### Comment · Reviewer_cX41 · 2024-06-03
> > **response**
> >
> > Thank you for your response. I've read it and my assessment remains the same.

---

### Official Review · Reviewer_9SYA · 2024-05-12

**Rating:** 4
**Confidence:** 3
**Ethics Flag:** 1

**Summary:**

This paper proposes LAMPO, a method that uses LLMs to conduct few-shot ordinal classification by prompting the LLM to perform pairwise comparisons between the test example to be labeled and the few-shot examples available. They explore an array of ordinal classification tasks (e.g., SST, Yelp, etc.) using two LLMs (Palm2-Bison, Flan-T5-XXL).

Overall, the idea itself is simple and the results on the surface are promising, but there are aspects of the method and experimental methodology that could be further improved.

**Questions To Authors:**

Why do some of the baselines have such high variance in their results, e.g., 10-shot Offensive for Flan-T5? Is this because of some really poor runs?

**Reasons To Accept:**

**The method is simple, and provides an alternative to in-context learning for ordinal classification.** Because the method only works using pairwise comparisons, this means that input length limits are in general not a problem, which is not the case with in-context learning (ICL).

**Reasons To Reject:**

**The paper could explore more techniques for aggregating the results of the pairwise comparisons.** Currently, the pairwise comparisons are aggregated into a score number, and two methods are explored for obtaining class thresholds. However, the paper could be strengthened by exploring other ways of performing this aggregation, especially given the unique nature of pairwise data (e.g., one could cast this as an optimization problem, where the label that violates the smallest number of pairwise comparisons is set as the predicted class).

**One could argue that the central issue this paper tries to address—how to perform ICL when you are severely constrained in terms of input length—is less and less of a problem as newer LLMs support ever-larger context windows.** Repeating these experiments with more recent LLMs than the ones in the paper (e.g., Gemini, GPT-4, Llama-3, etc.) and showing the benefits remain when longer context windows are available would strengthen the paper’s central point.

**There are aspects of the experimental methodology that could be improved.**

- It is not clear whether it is fair to the baselines that if they cannot fit a 10-shot prompt in their context window, then the result should be “NA”; in this case, a practitioner would likely pare down the number of examples until it fits in that context window (e.g., using 7-shot, or a variable number by class). This is especially true of the Yelp-5 experiment: could you not do at least 1-shot in this case, for the baselines?
- In some cases the error bars around the results suggest that LAMPO might not be better than the baselines (e.g., PaLM SST-5, Lap14). It would be appropriate to discuss this, and you should highlight all models in a row that perform statistically equivalently.

---

> ### Author Rebuttal · Authors · 2024-05-30
>
> We appreciate your careful review and constructive feedback to help us improve our paper.
>
> - The paper could explore more techniques for aggregating the results of the pairwise comparisons.
>
> We appreciate your suggestion. As LAMPO is the first work along few-shot comparison-based ordinal classification, there are several future directions. Your suggestion is promising. Other directions may include intelligent selection of pairs and joint pairwise comparison in one LLM call.
>
> - One could argue that the central issue this paper tries to address—how to perform ICL when you are severely constrained in terms of input length—is less and less of a problem as newer LLMs support ever-larger context windows.
>
> This is a thoughtful point. We note that context length constraint is *one of* the major issues LAMPO wants to address. It is possible that long-context LLMs *may* have *more* severe position biases or still not optimal to make (the difficult) pointwise decision. These aspects may need further studies.
>
> - Questions about "NA" and error bars.
>
> We used “NA” mainly to show applicability of LAMPO, and did not mean to be unfair for baselines in terms of performance (they are missing, but not low). Per your suggestion, we performed 1-shot experiments on the Yelp-5 dataset on PaLM2:
>
> ICL: 41.47_{1.20}
>
> CC: NA
>
> GlobalE: 43.20_{1.47}
>
> LAMPO: 47.53_{1.40}
>
> The gap is noticeable, possibly because it is more difficult for the LLM to handle several long documents (for 1-shot 5 way, there are 6 long documents in the prompt). We will add more discussion and appreciate your suggestion.
>
> For error bars, we note that they are the variance from 3 runs and may not be appropriate for significance tests. We thus perform significance tests on the instance prediction level on PaLM2-S against the competitive GlobalE. On a two-tailed t-test at the p < 0.05 level, LAMPO is significantly better than GlobalE on all settings except for Offensive 10-shot.
>
> - Why do some of the baselines have such high variance in their results?
>
> Yes, we found that pointwise ICL can be fragile based on demonstrations. Our hypothesis is that some demonstrations can be harmful for performance. On the other hand, in LAMPO, each demonstration is compared separately, so a harmful demonstration only contributes 1/kn. We did not include this since we did not perform comprehensive study on this perspective, and will discuss more.

---

> > ### Comment · Reviewer_9SYA · 2024-06-01
> > **Rebuttal response**
> >
> > Thank you for your response!
> >
> > RE: Further exploration. Thank you for clarifying this point; however, I still think the paper in its current form could benefit from further exploration (as opposed to deferring it as future work).
> >
> > RE: Context length. I definitely agree that there may be other aspects at play here, and the paper could definitely be strengthened by exploring these and showing that such positional biases have an impact on this task (or citing relevant work that explores this).
> >
> > RE: 1-shot Yelp. Thank you for the additional experiments!
> > - Following up on this, is 1-shot the most that you can do given context length limits of the LLMs?
> > - To clarify, my point was broader that just Yelp: It was that, to be fair to the baselines, se should at least do k-shot for the highest k possible given context length limits (1-shot being the minimum).

---

> > > ### Author Response · Authors · 2024-06-03
> > > **We appreciate your constructive feedback**
> > >
> > > We again thank the reviewer for their constructive feedback to help us improving the paper.
> > >
> > > Regarding the last point: We appreciate your explanation and agree it is a great suggestion. Yes, 1-shot (5+1 examples in the input prompt) is the headroom for Yelp-5 on the LLMs we use. Note that we follow the publicly hosted benchmark which provides up to 10-shot for each dataset. We will try to add ablation studies to go beyond 10-shots and update here or in the next version, which needs nontrivial efforts including gathering and sampling additional training demonstrations and performing inference. We feel the benefit of LAMPO may be more prominent as position bias can be more difficult to mitigate (due to its combinatorial nature) for existing baselines such as GlobalE. We really appreciate the suggestion.

---

### Official Review · Reviewer_jhug · 2024-05-15

**Rating:** 7
**Confidence:** 3
**Ethics Flag:** 1

**Summary:**

This article presents the LAMPO approach to performing sentiment analysis.
This approach is "LLM free" which means one can use any LLM they want since the strategy focuses on a way to compare scores extracted in two different ways, then the approach compares the scores and performs a decision process to extract the sentiment level.

**Questions To Authors:**

What are the performances of classical approaches against yours?
You discuss the ecological impact of your model. Could you elaborate in terms of energy and CO2 consumption?

**Reasons To Accept:**

The article is well presented and well written; the approach is well described.
The experiences sufficiently present the approach's effectiveness against classical prompting LLM approaches.

**Reasons To Reject:**

One can regret the approache is not compared to fine-tuned approach in order to have a benchmark.

---

> ### Author Rebuttal · Authors · 2024-05-30
>
> We appreciate your constructive feedback and advocate of our paper. Here we discuss some questions/concerns you raised.
>
> - One can regret the approache is not compared to fine-tuned approach in order to have a benchmark.
>
> Thank you for your suggestion. We agree a fine-tuned approach can serve as a good reference point. We note that this paper focused on fair comparative performance of different methods using the same LLMs. Comparison with fine-tuned models may have caveats such as potential data leakage. Nevertheless, we report fine-tuned numbers here using a T5 large model trained on each datasets’ respective training data (not available to LLMs method, despite potential data leakage). Below are the summarized numbers (note the best numbers can be from different LLMs and/or baselines):
>
> dataset, best baseline, best LAMPO, Fine-tuned
>
> twitter, 65.13, 66.87, 67.73
>
> SST-5, 54.8, 56, 56.80
>
> Yelp-5, NA, 51.13, 65.60
>
> Lap14, 77.23, 78.73, 78.60
>
> Hate, 58.84, 71.43, 46.94
>
> Offensive, 82.38, 81.10, 80.76
>
> Irony, 88.11, 89.61, 79.44
>
> We believe the results are appealing - on Lap14, Hate, Offensive, and Irony, LAMPO can outperform fine-tuned models. On twitter, SST-5, LAMPO can fill the gap by a large margin. Yelp-5 appears difficult for the LLMs we experimented with but LAMPO is the only one that can produce some reasonable results.
> We plan to add these to the appendix for the future version and discuss the potential caveats. We appreciate your suggestions.
>
>
> - You discuss the ecological impact of your model. Could you elaborate in terms of energy and CO2 consumption?
>
> This is a great question. Overall we believe that, although LAMPO proposes a paradigm shift for ordinal classification, it is still comparable to existing methods in terms of how LLMs are used, so it should not lead to noticeably more energy cost or CO2 consumption.

---

> > ### Comment · Reviewer_jhug · 2024-06-03
> > **Feedbacks**
> >
> > Hello,
> > thanks for your answers and precisions.
> > Best,

---

### Decision · Program_Chairs · 2024-07-10

**Decision:**

Accept

**Comment:**

The paper introduces a method to use LLMs for solving few-shot multi-class ordinal classification tasks. Compared with conventional ICL, the proposed method aggregates the relative comparative decisions between the test instance and each demonstration. The cost becomes larger while it overcomes some limitations such as ordering biases. One out of four reviewers tends to reject the submission because of the lack of comparisons with more techniques for aggregating the results of the pairwise comparisons, and LLMs with larger context window. The experimental settings can also be improved as suggested.